

# The comparative plastisphere microbial community profile at Kung Wiman beach unveils potential plastic-specific degrading microorganisms

Nutsuda Chaimusik[1], Natthaphong Sombuttra[1],
Yeampon Nakaramontri[2], Penjai Sompongchaiyakul[3],
Chawalit Charoenpong[3], Bungonsiri Intra[4,5] and Jirayut Euanorasetr[1]

[1] Laboratory of Biotechnological Research for Energy and Bioactive Compounds, Department of Microbiology, Faculty of Science, King Mongkut's University of Technology Thonburi, Bangkok, Thailand
[2] Sustainable Polymer & Innovative Composites Material Research Group, Department of Chemistry, Faculty of Science, King Mongkut's University of Technology Thonburi, Bangkok, Thailand
[3] Department of Marine Science, Faculty of Science, Chulalongkorn University, Bangkok, Thailand
[4] Department of Biotechnology, Faculty of Science, Mahidol University, Bangkok, Thailand
[5] Mahidol University and Osaka University Collaborative Research Center for Bioscience and Biotechnology, Faculty of Science, Mahidol University, Bangkok, Thailand

Corresponding author
Jirayut Euanorasetr,
jirayut.eua@kmutt.ac.th

## ABSTRACT

**Background:** Plastic waste is a global environmental issue that impacts the well-being of humans, animals, plants, and microorganisms. Microplastic contamination has been previously reported at Kung Wiman Beach, located in Chanthaburi province along with the Eastern Gulf of Thailand. Our research aimed to study the microbial population of the sand and plastisphere and isolate microorganisms with potential plastic degradation activity.

**Methods:** Plastic and sand samples were collected from Kung Wiman Beach for microbial isolation on agar plates. The plastic samples were identified by Fourier-transform infrared spectroscopy. Plastic degradation properties were evaluated by observing the halo zone on mineral salts medium (MSM) supplemented with emulsified plastics, including polystyrene (PS), polylactic acid (PLA), polyvinyl chloride (PVC), and bis (2-hydroxyethyl) terephthalate (BHET). Bacteria and fungi were identified by analyzing nucleotide sequence analysis of the 16S rRNA and internal transcribed spacer (ITS) regions, respectively. 16S and ITS microbiomes analysis was conducted on the total DNA extracted from each sample to assess the microbial communities.

**Results:** Of 16 plastic samples, five were identified as polypropylene (PP), four as polystyrene (PS), four as polyethylene terephthalate (PET), two as high-density polyethylene (HDPE), and one sample remained unidentified. Only 27 bacterial and 38 fungal isolates were found to have the ability to degrade PLA or BHET on MSM agar. However, none showed degradation capabilities for PS or PVC on MSM agar. Notably, *Planococcus* sp. PP5 showed the highest hydrolysis capacity of 1.64 ± 0.12. The 16S rRNA analysis revealed 13 bacterial genera, with seven showing plastic degradation abilities: *Salipiger*, *Planococcus*, *Psychrobacter*, *Shewanella*, *Jonesia*,

*Bacillus*, and *Kocuria*. This study reports, for the first time of the BHET-degrading properties of the genera *Planococcus* and *Jonesia*. Additionally, The ITS analysis identified nine fungal genera, five of which demonstrated plastic degradation abilities: *Aspergillus*, *Penicillium*, *Peacilomyces*, *Absidia*, and *Cochliobolus*. Microbial community composition analysis and linear discriminant analysis effect size revealed certain dominant microbial groups in the plastic and sand samples that were absent under culture-dependent conditions. Furthermore, 16S and ITS amplicon microbiome analysis revealed microbial groups were significantly different in the plastic and sand samples collected.

**Conclusions:** We reported on the microbial communities found on the plastisphere at Kung Wiman Beach and isolated and identified microbes with the capacity to degrade PLA and BHET.

## INTRODUCTION

The plastics industry has experienced rapid growth since its emergence in human daily life only 100 years ago. In 2021, global plastic production reached a staggering 390 million tons, which is expected to continue increasing. Among the various types of plastic, polyvinyl chloride (PVC), polyethylene (PE), polypropylene (PP), polystyrene (PS), and PE terephthalate (PET) constitute >80% of the total plastic content (*Urbanek, Rymowicz & Mirończuk, 2018*).

Plastic waste has become a pressing environmental issue affecting both human well-being and biodiversity, including animals, plants, and microorganisms that play an important role as decomposers in ecosystems. It is estimated that >4.8 million tons of plastic are dumped annually into the oceans (*Boucher & Friot, 2017*). The Thailand Development Research Institute has reported that at least 1.03 tons of plastic wastes are annually discarded into Thai waters. Due to the extensive use of plastics in human activity, debris is often washed and transported into the ocean, leading to microplastic contamination in the ecosystem. Microplastics, which are plastic particles <5 mm, have been found to accumulate in aquatic animals and ecosystems, such as bivalves on the Eastern Coast of Thailand (*Tharamon, Praisanklul & Leadprathom, 2016*), marine invertebrates in Southeast Asia ingest microplastics (*Hasbudin, Harith & Idrus, 2022*). Microplastics have also been found on the Eastern Gulf of Thailand, including Kung Wiman Beach in Chanthaburi province (*Bissen & Chawchai, 2020*), on the Western coast of Thailand, including Ban don Bay in Surat Thani province (*Chinfak et al., 2021*), and Bang Yai canal mouth, Phuket province (*Jiwarungrueangkul et al., 2021*). The accumulation of microplastics in marine biota and seawater serves as an indicator of ecosystem contamination, which not only cause pollution, but also impact human health.

Exposure to microplastics can cause tissue damage, oxidative stress, and alterations in immune-related gene expression in fish (*Tharamon, Praisanklul & Leadprathom, 2016*). Microplastics also act as carriers for toxic substances such as polycyclic aromatic hydrocarbons, polychlorinated biphenyls, dichlorodiphenyltrichloroethane, and dioxins, some of which are known to have carcinogenic properties in humans (*Bhuyan, 2022*). While research on the specific effects of microplastics on human health is ongoing, it is crucial to monitor the extent of microplastic contamination in water sources and aquatic organisms.

Various methods can be used to degrade plastic, such as UV light, heat, chemicals, or biodegradation using microorganisms that degrade plastic into carbon dioxide and water. Some microorganisms involved in plastic degradation produce various hydrolytic enzymes, including cutinase, lipase, protease, esterase, laccase, or peroxidase (*Srikanth et al., 2022*). For example, PETases from *Ideonella sakaiensis* 201-F6 can degrade PET into terephthalic acid (TPA) and ethylene glycol, which are environmentally friendly byproducts (*Yoshida et al., 2016*), cutinases from *Thermobifida alba* AHK119 for PLA, PET, and polybutylene succinate degradation, and lipases from *Thermomyces lanuginosus* for polycaprolactone (PCL) degradation (*Zhou et al., 2022*). Additionally, Arctic Sea ice bacteria such as *Marinomonas*, *Pseudoalteromonas*, and *Pseudomonas* exhibit extracellular lipase activity, which can hydrolyze polyesters like PCL (*Urbanek, Rymowicz & Mirończuk, 2018*). The laccase from *Rhodococcus ruber* C208 can degrade LDPE films. Moreover, the esterase from *Enterobacter* sp. HY1 can degrade BHET (*Qiu et al., 2020*).

Similarly, other studies have identified fungal enzymes involved in plastic degradation. For example, the cutinase from *Fusarium oxysporum* could degrade PET (*Zhou et al., 2022*), the lipase B from *Candida antarctica* could effectively hydrolyze PET into TPA, the laccase from *Cochliobolus* sp. could degrade low molecular weight PVC (*Sumathi et al., 2016*). Additionally, the esterase from *Curvularia senegalensis* has been reported to degrade poly (butylene succinate-co-adipate) and PU. Also, the esterase from *Penicillium griseofulvum* could degrade PU. Moreover, the manganese peroxidases and lignin peroxidases from *Aspergillus flavus*, *A. niger*, and *Fusarium graminearum* could biodegrade PE carrier bags, with *A. niger* and *A. flavus* producing the most lignin peroxides (*Srikanth et al., 2022*).

Studies on the microbial communities residing on plastic debris, known as the plastisphere, are expanding due to the availability of high-throughput DNA sequencing (*Zettler, Mincer & Amaral-Zettler, 2013*). With exogenous and hydrophobic substrates, the plastic surface provides a unique niche for the growth and proliferation of diverse microorganisms.

Therefore, this research focuses on studying the marine microbial diversity of microplastic-contaminated sites at Kung Wiman Beach by conventional isolation and amplicon sequencing of the 16S rRNA and internal transcribed spacer (ITS) region. This study aims to reveal the microbiota in microplastic-contaminated marine environments and identify microorganisms with the potential to degrade plastics.

## MATERIALS AND METHODS

### Collection of seawater, sand, and plastics samples

Sixteen samples of plastic debris were collected from the beach surface. Togethering with, four sand samples were collected from four different locations on Kung Wiman Beach according to Fig. S1. All were kept in zip lock plastic bag. For microbiome analysis, one-gram samples of plastics and sand were preserved in ampoules filled with DNA/RNA Shield solution (Zymo Research, Tustin, CA, USA) and stored at 4 °C. Seawater was collected from the near-shore area and stored in a gallon bottle at room temperature for media preparation, dilution, and salinity experiment. Then, all samples were transported to the laboratory in an iced container.

### Determination of the physical properties of sand and seawater

Four sand samples were incubated at 60 °C for 24 h until completely dry. Next, approximately 500 g of the dried sand was sifted through a sieve shaker with six sieves of different sizes (63, 125, 250, 500 μm, 1, and 2 mm) for 10 min. Then, the sand samples remaining on each sieve were weighed, and the particle size fraction was calculated.

The organic carbon content of the sand particles was determined using the modified Walkley–Black method (*Loring & Rantala, 1992*). The dried sand samples were sifted through a 2 mm sieve and then finely and homogeneously crushed using an agate mortar. The accuracy of the organic carbon content in the sand samples was evaluated using dextrose ($C_6H_{12}O_6$) as the standard. It is important to note that 0.01 g of dextrose contains 39.99% carbon.

The pH and salinity measurements were conducted using a pH meter and salinity refractometer, respectively. The specific gravity was also calculated.

### Plastics type analysis by Fourier-transform infrared (FT-IR) spectroscopy

Sixteen plastic samples were categorized based on their collection location. The plastic types were examined by attenuated total reflectance FT-IR (ATR-FTIR) using a Nicolet 6700 FT-IR spectrometer (Thermo Fisher Scientific, Waltham, MA, USA). The analysis was conducted within a wave number range of 650–4,000 $cm^{-1}$, with a resolution of 32 $cm^{-1}$. Each sample was analyzed 32 times. The obtained spectra were compared with the standard peaks of various plastics (*Jung et al., 2018*).

### Culture-dependent method

Microorganisms were extracted from the samples using the ultrasonic method. Plastic samples were cut into three–four small pieces, placed into 10 mL of sterile seawater, and vortexed for 1 min. The extraction of microorganisms from the plastic sheet was performed in an Ultrasonic Cleaner E100H (Elma, Singen, Germany) for 5 min, followed by vortexing for 1 min. For the sand samples, approximately 1 g or 1 spatula of sand was added to 9 mL of sterile seawater and vortexed for 1 min. Therefore, a microbial solution

was obtained from each plastic and sand sample. A total of 20 samples of the extracted microbial solution underwent ten-fold serial dilution and 100 μl of the microbial solution were spread on marine agar (MA) supplemented with cycloheximide to inhibit fungal growth and potato dextrose agar (PDA) supplemented with nalidixic acid to inhibit bacteria growth. The cultured microorganisms were incubated at room temperature for 1 month. Then, the distinct colony characteristics of microorganisms from each sample were collected and counted, and the number of microbial colonies (colony-forming unit (CFU)/mL) was calculated.

## 16S rRNA/ITS gene amplification and identification

The genomic DNA from the microorganism was extracted by the microwave method with adapted condition from the previous study (*Dashti et al., 2009*). Bacterial colonies were dissolved in 500 μl of 1X TE buffer and were heated for 10 s by microwave oven. After centrifugation for 5 min at 13,000 rpm, the supernatant was used for the template of PCR. Amplification reactions were prepared to final volume of 20 μl containing 2X PCR Master mix solutions (*i-Taq*$^{TM}$; iNtRON Biotechnology, Seongnam, Korea), 10 pmols of forward and reverse primer, and template DNA. For bacteria, the 16S rRNA region was polymerase chain reaction (PCR)-amplified using the primers 27F (5′-AGAGTTTGATCMT GGCTCAG-3′) and 1492R (5′-TACGGYTACCTTGTTACGACTT-3′). For fungi, the ITS region was PCR-amplified using the primers ITS4 (5′-TCCTCCGCTTATTGATATGC-3′) and ITS5 (5′-GGAAGTAAAAGTCGTAACAAGG-3′). The PCR conditions consisted of an initial denaturation at 95 °C for 5 min, followed by 30 cycles involving denaturation at 95 °C for 30 s, annealing at 52 °C for 30 s, and extension at 72 °C for 30 s. The process concluded with a final extension step at 72 °C for 5 min. The PCR products were verified through agarose gel electrophoresis and subsequently purified using the AccuPrep PCR Purification Kit (Bioneer, Daejeon, Korea). The purified PCR products were subjected to Sanger sequencing, performed by U2Bio Company (Bangkok, Thailand). The resulting partial 16S rRNA and ITS sequences had approximate lengths of 1,500 and 700 bp, respectively. The bacterial and fungal sequences were identified using EzBiocloud (*Yoon et al., 2017*) and UNITE (*Kõljalg et al., 2013*), respectively. The nucleotide sequences have been deposited in the GenBank database under accession numbers OR185572–OR185595 (bacteria) and OR205218–OR205249 (fungi).

A phylogenetic tree was constructed using MEGA11 software (*Tamura, Stecher & Kumar, 2021*). The evolutionary history was inferred using the neighbor-joining method (*Saitou & Nei, 1987*). The bootstrap consensus tree inferred from 1,000 replicates was taken to represent the evolutionary history of the analyzed taxa. Branches corresponding to partitions reproduced in <50% of bootstrap replicates are collapsed. The evolutionary distances were computed from 58 nucleotide sequences using the maximum composite likelihood method and were in the units of the number of base substitutions per site. The analysis included all codon positions (1st, 2nd, 3rd, and noncoding). All ambiguous positions were removed for each sequence pair (pairwise deletion option). The final dataset comprised 852 positions.

## Preparation of MSM agar with emulsified plastic

Plastic degradation properties were tested on mineral salts medium (MSM) agar supplemented with 0.5% emulsified plastic: PS (Sigma-Aldrich, St. Louis, MO, USA), PVC (Sigma-Aldrich, St. Louis, MO, USA), PLA (Goodfellow, England), or bis (2-hydroxyethyl) terephthalate (BHET; Sigma-Aldrich, St. Louis, MO, USA). Two grams of each plastic was put into a 250 mL flask and dissolved with 60 mL of dichloromethane (Honeywell, Charlotte, NC, USA) and heated at 55 °C and agitated with a magnetic stirrer before 15 mL of 2% sarkosyl (Wako Pure Chemical, Osaka, Japan) was added. For BHET was dissolved with dimethyl sulfoxide (Thermo Fisher Scientific, Waltham, MA, USA) and heated only. The plastic mixtures were processed with a VC/VCX 750 High-Intensity Ultrasonic Processor (Sonics, Oklahoma City, OK, USA) at 60% amplitude, 20 s run time, and 10 s rest time for a total time of 10 min. Next, 100 mL of distilled water was added to each plastic solution and mixed for 10 min until it turned milky white. Then, the plastic solution was adjusted in volume to 400 mL with distilled water to obtain a 0.5% plastic emulsion for all four plastic types.

The MSM agar was prepared by adding 0.5 g of yeast extract, 2 g of $(NH_4)_2SO_4$, 1 g of $K_2HPO_4$, 0.5 g of $KH_2PO_4$, 0.25 g of $MgSO_4.7H_2O$, and 7.5 g of agar in 400 mL of distilled water. MSM agar supplemented with PS or PVC was prepared by adding 100 mL of emulsified PS or PVC to the 400 mL of MSM before sterilization in an autoclave. MSM agar supplemented with PLA or BHET was prepared by adding 100 mL of emulsified PLA or BHET plastic to the 400 mL of MSM after sterilization in an autoclave.

## Plastic degrading properties test

Each isolated bacterium/fungus colony was tested on MSM agar supplemented with each emulsified plastic. Bacterial colonies were streaked on the agar, while fungal colonies were cut off and placed on the agar. The plate was further incubated at room temperature for 1 month, then the hydrolysis capacity (HC) was calculated as follows:

$$HC \text{ value } = \frac{\text{diameter of the clear zone around the bacterial colony}}{\text{diameter of the bacterial colony}}$$

## 16S and ITS amplicon sequencing

Four plastic debris from each plastic type (PS, PP, PET) and four sand samples from Kung Wiman Beach were performed in 16S and ITS amplicon sequencing analysis. These samples were processed and analyzed by Zymo Research Corporation using the ZymoBIOMICS Targeted Sequencing Service (Zymo Research, Tustin, CA, USA). The ZymoBIOMICS-96 MagBead DNA Kit (Zymo Research, Tustin, CA, USA) was used to extract DNA using an automated platform. The targeted bacterial 16S ribosomal RNA sequencing library was prepared using the Quick-16S NGS Library Prep Kit (Zymo Research, Tustin, CA, USA); the 16S primers (341F; 5′-CCTACGGGDGGCWGCAG-3′ and 806R; 5′-GACTACNVGGGTMTCTAATCC-3′) amplified the V3–V4 region of the 16S rRNA gene. The targeted fungal ITS sequencing library was also prepared using the Quick-16S NGS Library Prep Kit but with the 16S primers replaced by custom ITS2

primers (ITS3-F; 5′-GCATCGATGAAGAACGCAGC-3′ and ITS4-R; 5′-TCCTCCGCTTATTGATATGC-3′). The sequencing library was generated through a novel library preparation process, wherein PCR reactions were performed in real-time PCR machines to control cycles and, consequently, limit the formation of PCR chimeras. Quantification of the final PCR products was performed using quantitative PCR fluorescence readings, and the products were combined based on equal molarity. The resulting pooled library underwent purification using the Select-a-Size DNA Clean & Concentrator Kit (Zymo Research, Tustin, CA, USA) and was subsequently quantified using TapeStation (Agilent Technologies, Santa Clara, CA, USA) and Qubit (Thermo Fisher Scientific, Waltham, MA, USA). The ZymoBIOMICS Microbial Community Standard (Zymo Research, Tustin, CA, USA) served as a positive control for each DNA extraction. Additionally, the ZymoBIOMICS Microbial Community DNA Standard (Zymo Research, Tustin, CA, USA) was employed as a positive control for each specific library preparation. Negative controls were included to evaluate the bioburden level throughout the wet-lab process. The final library was subjected to sequencing on an Illumina MiSeq system, using the v3 reagent kit (600 cycles) along with a 10% PhiX spike-in. The 16S and ITS amplicon sequencing raw sequence reads have been deposited in the GenBank database under accession numbers PRJNA997562.

## Microbiome analyses

Unique amplicon sequence variants were identified from raw reads through the DADA2 pipeline (*Callahan et al., 2016*) by Zymo Research Corporation. The DADA2 pipeline was also employed to eliminate potential sequencing errors and chimeric sequences. Taxonomy assignment was carried out using Uclust from QIIME (v.1.9.1) with the Zymo Research Database, a 16S database, or ITS internally designed and curated as a reference. Composition visualization, alpha-diversity, and beta-diversity analyses were conducted with QIIME (v.1.9.1) (*Caporaso et al., 2010*). In cases where applicable, taxonomies exhibiting significant abundances in different groups were identified through linear discriminant analysis effect size (LEfSe) (*Segata et al., 2011*) using default settings. Other analyses were performed with internal scripts, including heatmaps, Taxa2ASV Decomposer, and principal coordinates analysis (PCoA) plots. To re-analyze non-classified fungal sequences, the raw ITS sequence reads in FASTQ format (provided by Zymo Research Corporation, Tustin, CA, USA) were further processed using the DADA2 pipeline to generate amplicon sequence variant (ASV). The total ITS amplicon sequences in each sample were determined using QIIME 2 (v.2022.11). Subsequently, the consensus sequences from each ASV were analyzed using the BLAST tool against the GenBank database (*Altschul et al., 1990*).

The statistical analysis was conducted by our researcher group using QIIME 2 (v.2022.11). Alpha diversity, including Shannon indices, was assessed using Kruskal-Wallis rank-sum tests to evaluate differences in microbial diversity among each sample. Principal Coordinates Analysis (PCoA) based on Bray-Curtis dissimilarities was employed to visualize beta diversity, assessing variations in microbial diversity between samples, and permutational multivariate analysis of variance (PERMANOVA) was employed to assess

statistical significance. A *p*-value of <0.05 was considered statistically significant in all statistical experiments.

## RESULTS

### Sand, seawater, and plastics type analysis

The illustration showing the experimental design was shown in Fig. 1. Twenty samples of different plastic types and sand were collected at four different locations around Kung Wiman Beach in Chanthaburi Province, Thailand (Fig. S1): GPS point 1 = 12°60′29.50′N 101°87′71.24′E; GPS point 2 = 12°60′33.29″N 101°87′68.61″E; GPS point 3 = 12°60′45.31″ N 101°87′51.91″; and GPS point 4 = 12°60′33.43″N 101°87′68.62″E.

The particle size of the four sand samples was mostly fine sand (125–250 μm) and medium sand (250–500 μm), with the organic carbon content between 0.04% and 0.09% (Fig. S2 and Table S1). The seawater's pH was 7.63 and its salinity was 29 psu.

Sixteen plastic samples (Table S2) were identified by ATR-FTIR. By comparing the plastic samples' ATR-FTIR peaks with the standard peaks of various plastics as follows: Polypropylene (2,950, 2,915, 2,838, 1,455, 1,377, 1,166, 972, 840, and 808 cm$^{-1}$), Polystyrene (3,024, 2,847, 1,601, 1,492, 1,451, 1,027, and 694 cm$^{-1}$), Polyethylene terephthalate (1,713, 1,241, 1,094, and 720 cm$^{-1}$), and high-density PE (2,915, 2,845, 1,462, 730, and 717 cm$^{-1}$), revealed that five samples were PP (sample codes: 1.3, 1.4, 4.1, 4.2, and 4.4), four were PS (sample codes: 2.1, 2.2, 2.3, and 2.4), four were PET (sample codes: 3.1, 3.2, 3.4, and 4.3), two were HDPE (sample codes: 1.2, and 3.3), and one was an unidentified plastic (sample code: 1.1) (Fig. S3).

### Isolation of microorganisms using culture-dependent methods

Fifty-four and 62 isolates of bacteria and fungi were successfully isolated from plastic and sand samples on MA supplemented with cycloheximide and PDA supplemented with nalidixic acid, respectively. The bacterial colonies ranged from $2.5 \times 10^3$–$5.10 \times 10^7$ CFU/g, while fungal colonies were <$2.5 \times 10^2$ CFU/g.

After checking viability during cultivation, this study selected only 27 bacterial and 38 fungal isolates, based on distinct colony characteristic and the source of isolation (PP, PS, PET, and sand samples). These bacterial isolates were categorized into four groups as follows: PP plastic (eight isolates), PS plastic (six isolates), PET plastic (eight isolates), and sand (five isolates). In case of fungal isolates, they were categorized as follows: PP plastic (ten isolates), PS plastic (nine isolates), PET plastic (eleven isolates), and sand (eight isolates). The strain codes assigned to the bacteria and fungi in each isolate are shown in Table 1.

The 16S and ITS regions were successfully molecularly classified. Thirteen bacterial genera were identified through 16S rRNA analysis with EzBiocloud: *Marinobacter*, *Pseudoalteromonas*, *Salipiger*, *Planococcus*, *Psychrobacter*, *Shewanella*, *Exiguobacterium*, *Marinomonas*, *Cobetia*, *Jonesia*, *Bacillus*, *Kocuria*, and *Halomonas*. The isolated fungi were classified into nine genera based on their ITS region: *Aspergillus*, *Penicillium*, *Pestalotiopsis*, *Peacilomyces*, *Cochliobolus*, *Absidia*, *Fusarium*, *Diplodia*, and *Petromyces*

**Analysis of plastic using FT-IR**

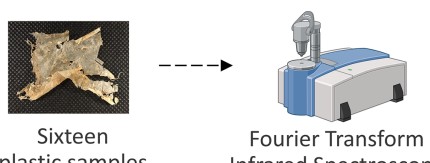

Sixteen plastic samples → Fourier Transform Infrared Spectroscopy

**Plastic and sand samples from Kung Wiman Beach, Thailand**

**16S/ITS amplicon sequencing**

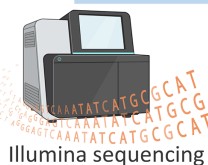

Illumina sequencing machine

- Amplicon sequencing using DADA2 pipelines
- Bioinformatic analysis by QIIME (v.1.9.1)

**Microbial isolation**

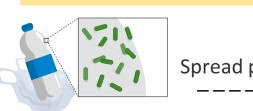 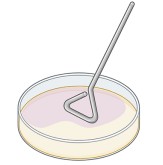

Extraction of microorganisms from the plastic and sand samples → Spread plate → MA+Cycloheximide and PDA+Nalidixic acid

Culture-dependent method / High throughput technology

Bacteria          Fungi

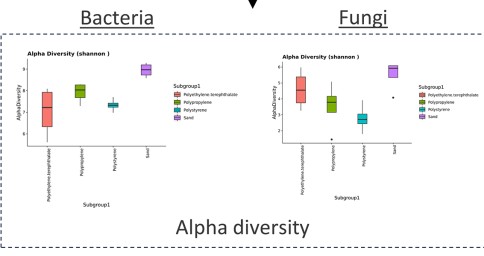

Alpha diversity

**Plastic-degrading microbes**

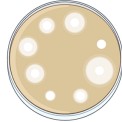

Testing on Mineral Salts Media (MSM) supplemented with PS/PVC/PLA/BHET plastic emulsion

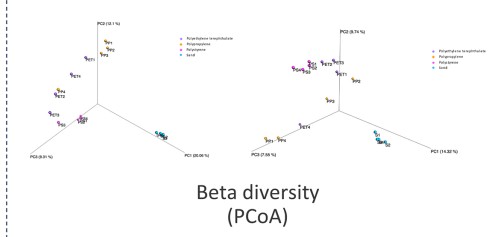

Beta diversity (PCoA)

**16S/ITS gene amplification and identification**

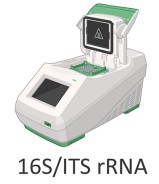 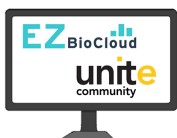

16S/ITS rRNA amplification → Sequencing → EZ BioCloud / unite community → Microbial identification

Microbial Community Composition

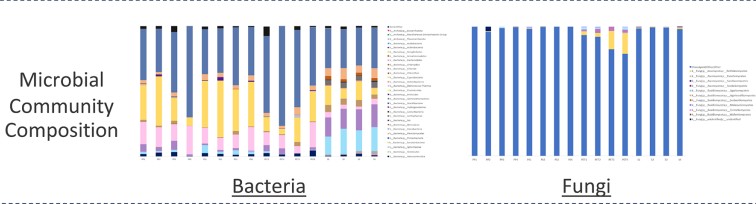

Bacteria          Fungi

**Figure 1** The experimental design. Created with BioRender.com.

**Table 1** The number and strain codes of bacterial and fungal isolates from three plastic types (PP, PS, and PET) and sand samples.

| Sample types | Sample codes | Bacterial isolates | | Fungal isolates | |
|---|---|---|---|---|---|
| | | Number | Strain code | Number | Strain code |
| PP | 1.3/1.4/4.1/4.2/4.4 | 8 | PP1–PP8 | 10 | PP1–PP10 |
| PS | 2.1/2.2/2.3/2.4 | 6 | PS1–PS6 | 11 | PS1–PS11 |
| PET | 3.1/3.2/3.4/4.3 | 8 | PET1–PET8 | 9 | PET1–PET9 |
| HDPE | 1.2/3.3 | – | – | – | – |
| Unidentified | 1.1 | – | – | – | – |
| Sand | 1/2/3/4 | 5 | Sand1–Sand5 | 8 | Sand1–Sand8 |
| Total | | 27 | | 38 | |

(A)

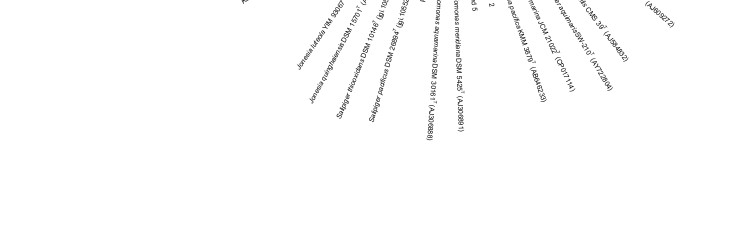

(B)

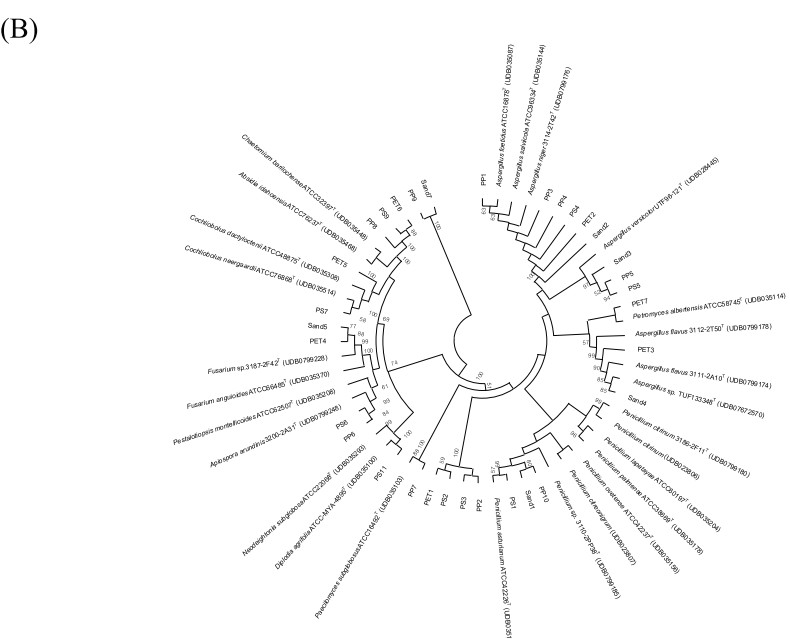

**Figure 2 Neighbor-joining phylogenetic trees were constructed using MEGA11 software. The percentage of replicate trees, in which the associated taxa clustered together in the bootstrap test (1,000 replicates), is indicated adjacent to the branches.** (A) 16S rRNA sequences, and (B) ITS sequences demonstrate the genetic relatedness of bacterial and fungal isolates, respectively, from plastisphere and sand samples.

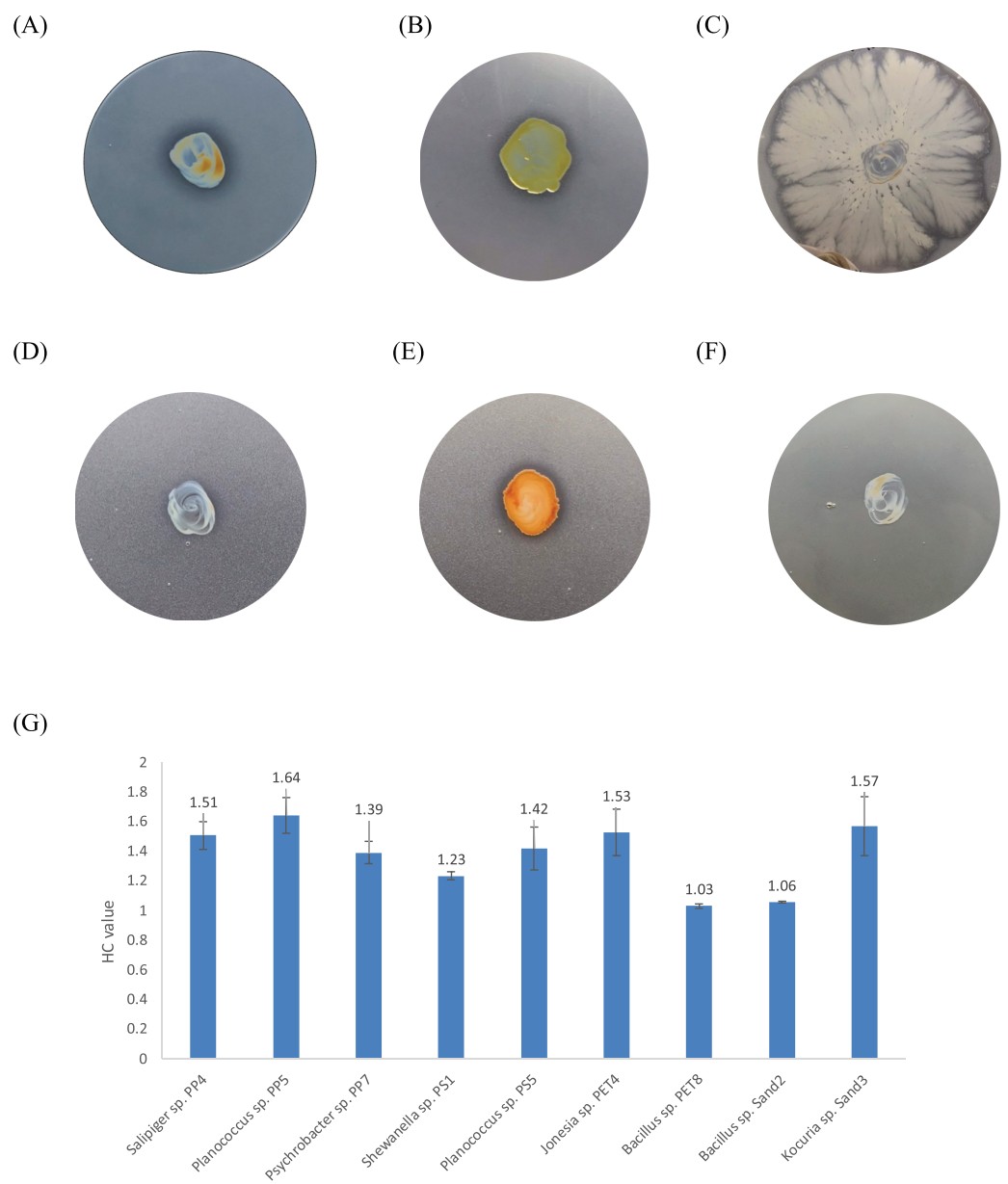

**Figure 3 The BHET degradation of bacterial isolates on MSM with emulsified BHET agar plates for 1 month.** (A) *Planococcus* sp. PP5, (B) *Jonesia* sp. PET4, (C) *Bacillus* sp. PET8, (D) *Psychrobacter* sp. Sand4, (E) *Rhodococcus ruber* DSM 45332 (the positive control), and (F) *Halomonas* sp. Sand5 (no clear zone). (G) The HC values of marine plastic-degrading bacterial isolates.

(Tables S3 and S4). The phylogenetic trees of the isolated bacteria and fungi are presented in Figs. 2A and 2B.

## Microorganism degradation on plastic-containing agar plates

One bacterium and 15 fungi exhibited clear zones on MSM agar containing emulsified PLA, indicating their ability to degrade this plastic. In addition, 14 bacteria and 19 fungi

(A) (B) (C)

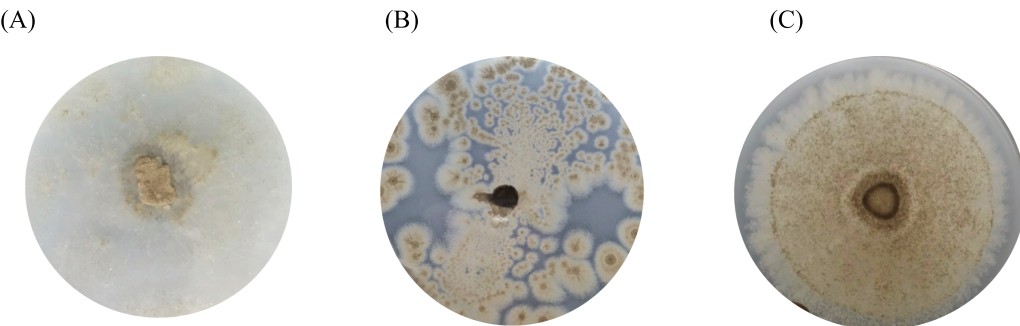

**Figure 4 The BHET degradation of fungal isolates on MSM with emulsified BHET agar plates for 1 month.** (A) *Absidia* sp. PS9, (B) *Penicillium* sp. PS1, and (C) *Aspergillus* sp. PET3.

demonstrated the ability to degrade BHET on MSM-emulsified plastic agar. However, none of the microorganisms demonstrated degradation of PS or PVC on MSM-emulsified plastic agar. The identification of bacteria and fungi through 16S rRNA and ITS region revealed the capability of seven bacterial genera (*Salipiger*, *Planococcus*, *Psychrobacter*, *Shewanella*, *Jonesia*, *Bacillus*, and *Kocuria*) and five fungal genera (*Aspergillus*, *Penicillium*, *Peacilomyces*, *Absidia*, and *Cochliobolus*) to degrade plastics. The results of clear zone are shown in Figs. 3A, 3B and 4. Additionally, the HC values of bacteria range from 1 to 1.64. Notably, *Planococcus* sp. PP5 obtained from a PP sample had the highest HC value of 1.64 ± 0.12, followed by *Salipiger* and *Kocuria* with HC values of 1.57 ± 0.18 and 1.55 ± 0.15, respectively (Table S3).

## Microbiome analysis

The 16S and ITS amplicon sequencing analysis in plastic and sand samples revealed the Shannon indices in Fig. 5, assessing the alpha diversity of the bacterial (Fig. 5A) and fungal (Fig. 5B) communities. Alpha diversity serves as a measure of the microbial diversity of each sample. The results indicated that the sand samples exhibited the highest alpha diversity for both bacteria and fungi, followed by PP, PS, and PET samples for bacteria and PET, PP, and PS samples for fungi. Additionally, the statistical analysis of Shannon indices for different groups was evaluated using the Kruskal-Wallis test, revealing significant differences in microbial communities among sample groups ($p$ value < 0.05).

Beta diversity assesses the variations in microbial diversity differences between samples. The principal coordinate analysis (PCoA) plot, as shown in Fig. 6, is generated using the matrix of pairwise distances between samples calculated by Bray-Curtis dissimilarity. It reveals distinctions in the communities of both bacteria and fungi between sand and plastic samples. Meanwhile, the microbial communities within plastic samples (PP, PS, and PET) exhibit similarities. The differences in beta-diversity were statistically assessed through permutational multivariate analysis of variance (PERMANOVA), as shown significant differences ($p$-value < 0.05) among sample groups.

The microbial community composition in Fig. 7 reveals that Proteobacteria were most abundant in PET samples (59.9%) and more commonly found in plastic samples than in sand samples, similar to Bacteroidetes. Cyanobacteria (36.3%) were more abundant in PS

(A)

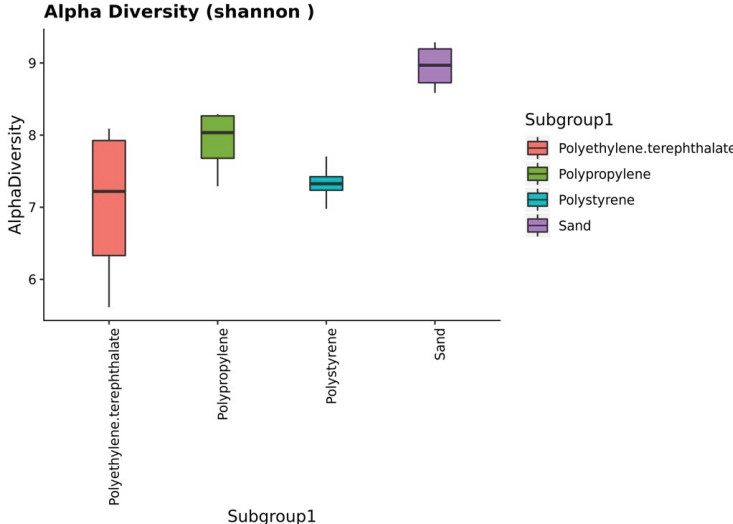

(B)

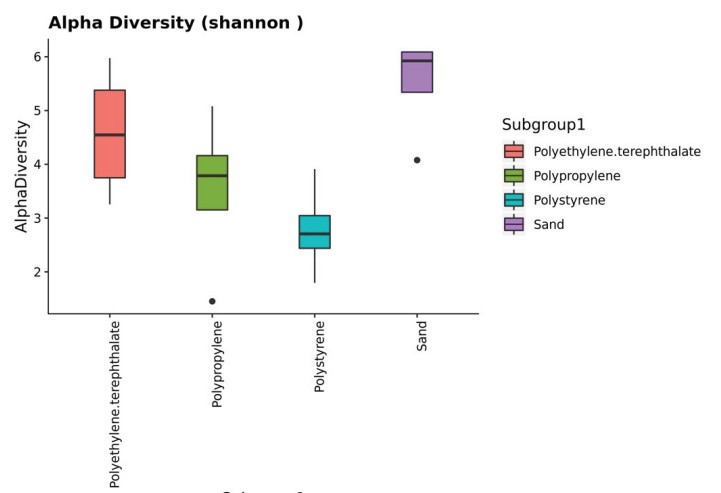

**Figure 5  The Shannon indices represent the alpha diversity of microbial isolates in PP, PS, PET, and sand samples.** (A) Bacteria and (B) fungi.               

samples than in other samples. On the other hand, Actinobacteria and Acidobacteria exhibited higher abundance in sand samples (15.9% and 16.9%, respectively) than in plastic samples. Regarding fungal communities, Ascomycota (11.6%) and Basidiomycota (2.8%) were the most prevalent and abundant groups in PET samples. However, a substantial portion (95.9%) of all samples could not be classified into specific fungal groups.

(A)

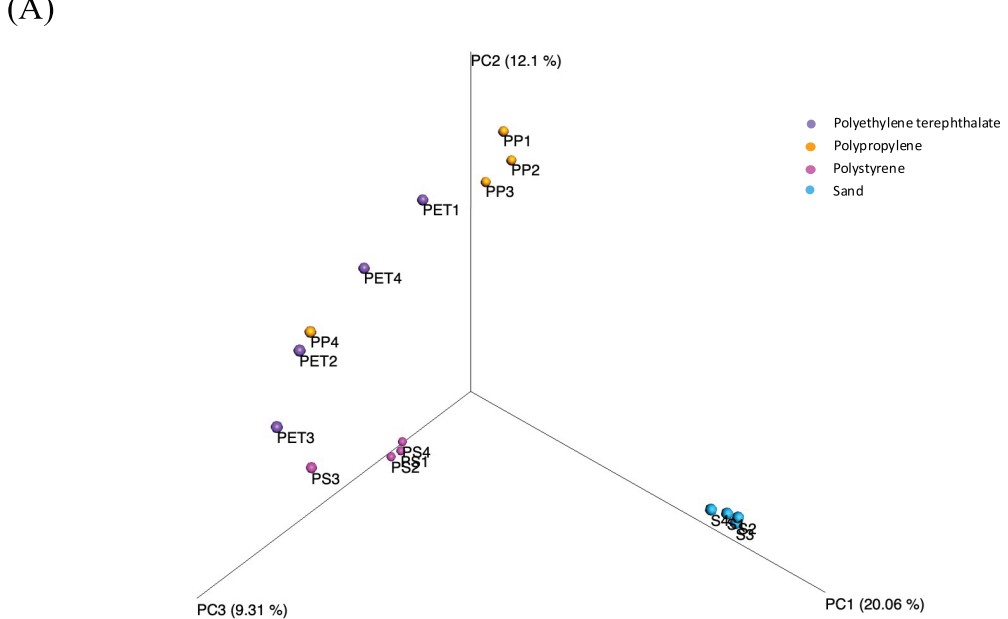

(B)

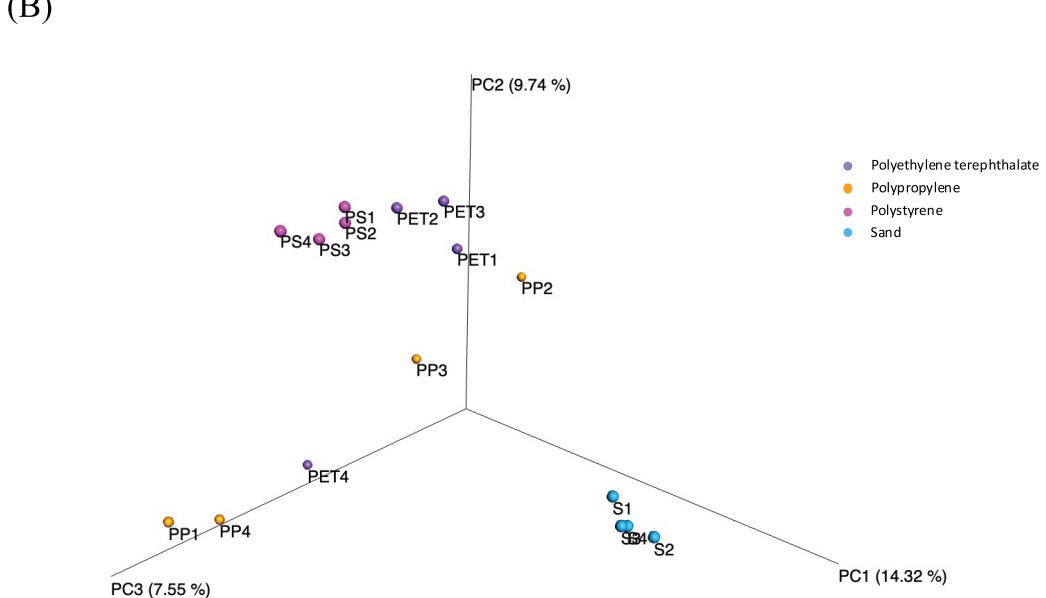

**Figure 6** **Principal coordinate analysis (PCoA) based on Bray-Curtis dissimilarities represents the beta diversity of microbial isolates between three plastic types (PP, PS, and PET) and sand samples.** (A) Bacteria and (B) fungi.

The fungal microbiome analysis conducted using the Zymo Research database revealed that 95.9% of the isolates could not be identified. Therefore, further analysis was performed using BLAST with the GenBank database. Blast results were obtained from nine samples

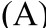

15/27

(A)

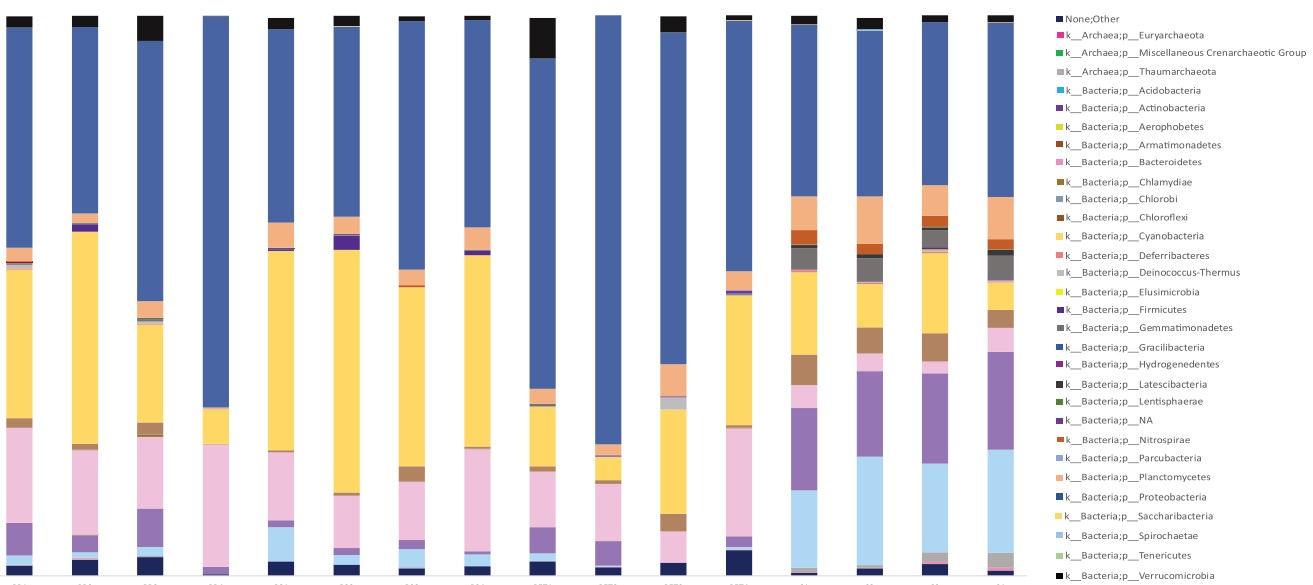

(B)

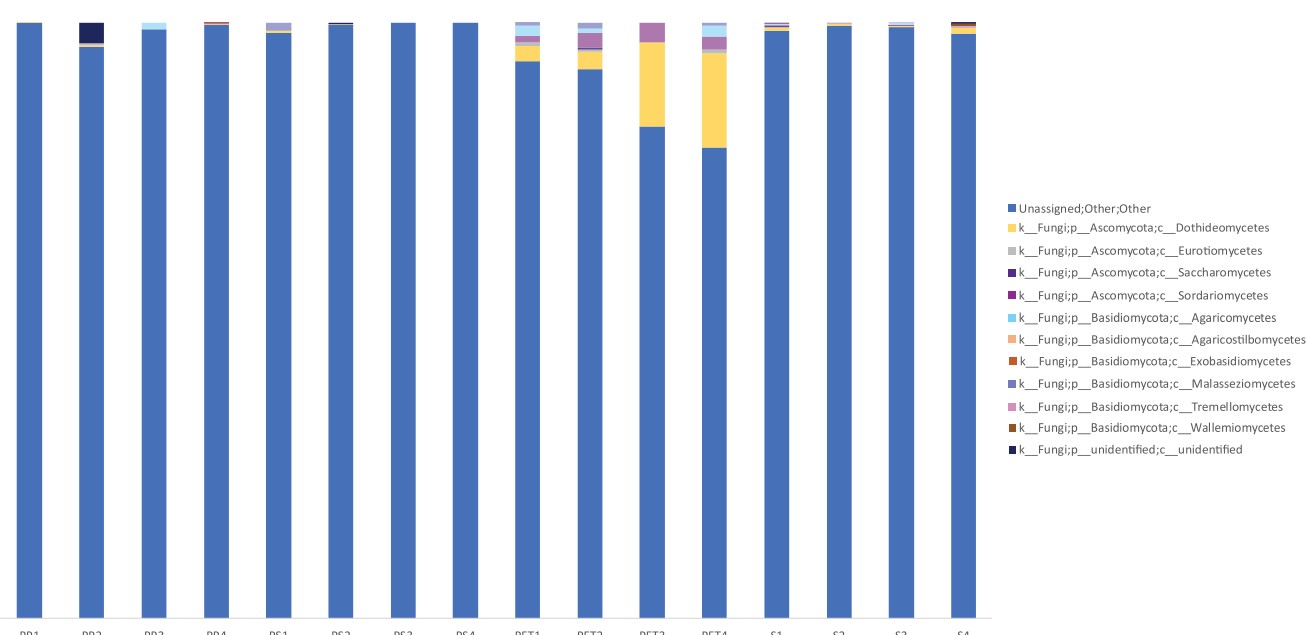

**Figure 7 Bar graph showing the microbial community composition of bacteria and fungi in PP, PS, PET, and sand samples.** (A) Bacteria at phylum level, and (B) fungi at class level.

(sample names: PP1, PP3, PP4, PS1, PS2, PS3, PS4, PET3, and PET4), while the other seven samples (sample names: Sand1, Sand2, Sand3, Sand4, PP2, PET1, and PET2) did not match any sequences in the database with the current search parameters. From the BLAST results shown in Fig. S4, the dominant group was identified as other eukaryotic groups (61.5%), including ciliates, diatoms, hydrozoans, crustaceans, cercozoans, anemones, tunicates, gastropods, bivalves, chytrids, sponges, green algae, and uncultured eukaryotes, while classified fungi were present in a smaller proportion (18.4%). The fungal class Dothideomycetes (9.23%) was the most abundant, followed by Sordariomycetes (3.08%) and Agaricomycetes (2.46%).

Linear discriminant analysis effect size (LeFSe) analysis helps to identify taxa whose distributions among different sample groups are statistically with $p$-value < 0.05 and the effect size (LDA score) higher than 2. The bacterial at genus level identified through LDA score, represent the most dominant microbial groups in each sample, revealing 15, 21, 15, and 22 bacterial genera in the PP, PS, PET, and sand samples, respectively (Fig. 8A). Similarly, the fungal at genus level identified through microbial composition bar plot, with results showing 7, 3, 26, and 19 fungal genera in the PP, PS, PET, and sand samples, respectively (Fig. 8B). On the other hand, bacterial groups isolated under culture-dependent conditions represented 5, 4, 6, and 5 genera in the PP, PS, PET, and sand samples, respectively (Fig. 9A), while isolated fungal groups represented 5, 6, 4, and 3 genera in the PP, PS, PET, and sand samples, respectively (Fig. 9B). Moreover, the 16S microbiome analysis of plastic and sand samples revealed the dominant bacterial genera that were absent in the culture-dependent conditions, such as *Prochlorococcus* in the PP samples, *Ruegeria* in the PS samples, *Erythrobacter* in the PET samples, and *Rhodopirellula* in the sand samples. The ITS microbiome indicated that *Cladosporium* and *Malassezia* were the most common fungal genera, while the fungi groups under culture-dependent conditions were mostly *Aspergillus* and *Penicillium*. Moreover, six bacterial genera (*Marinobacter*, *Pseudoalteromonas*, *Marinomonas*, *Jonesia*, *Bacillus*, and *Halomonas*) and two fungal genera (*Aspergillus* and *Penicillium*) were present in both isolation and microbiome analyses. Furthermore, comparing the microbial communities between plastic samples and sand samples, differences were observed in the bacterial communities of each sample. However, in the fungal group, some samples exhibited fungi from the same group, specifically *Cladosporium*, found in every sample. *Aspergillus* was identified in PP, PET, and sand samples; *Malassezia* was found in PS, PET, and sand samples; *Irpex* was present in PP and PET samples, and *Penicillium*, *Aureobasidium*, *Hortaea*, *Kalmusia*, and *Curvularia* were identified in PET and sand samples.

## DISCUSSION

Plastic waste is an environmental issue that is increasing yearly and leading to severe consequences, especially the microplastics contamination in the marine ecosystems, posing risks to both marine animals and human health. This study focused on screening and identifying microorganisms from Kung Wiman Beach in Chanthaburi Province, Thailand with the ability to degrade plastic. Microplastic accumulation has been observed in various provinces along the Eastern Gulf of Thailand, namely Chonburi, Rayong,

(A)

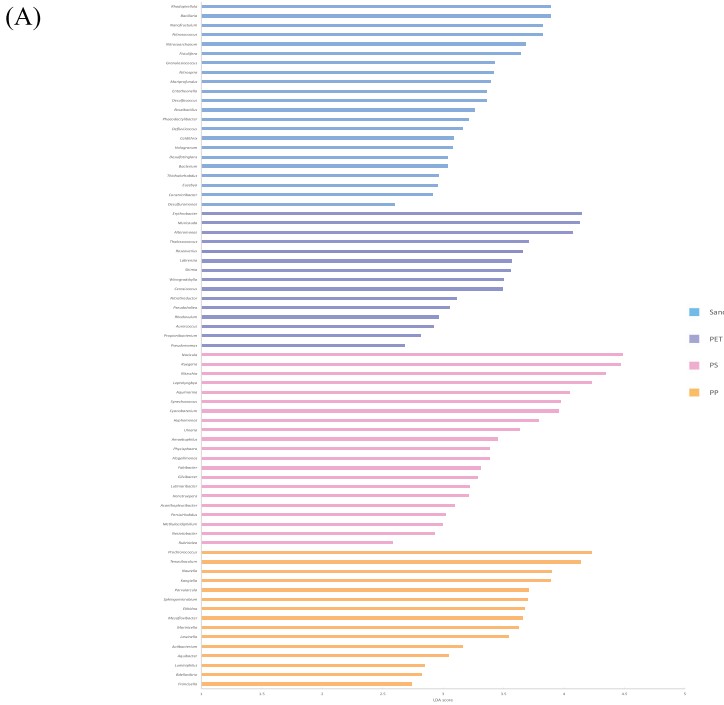

(B)

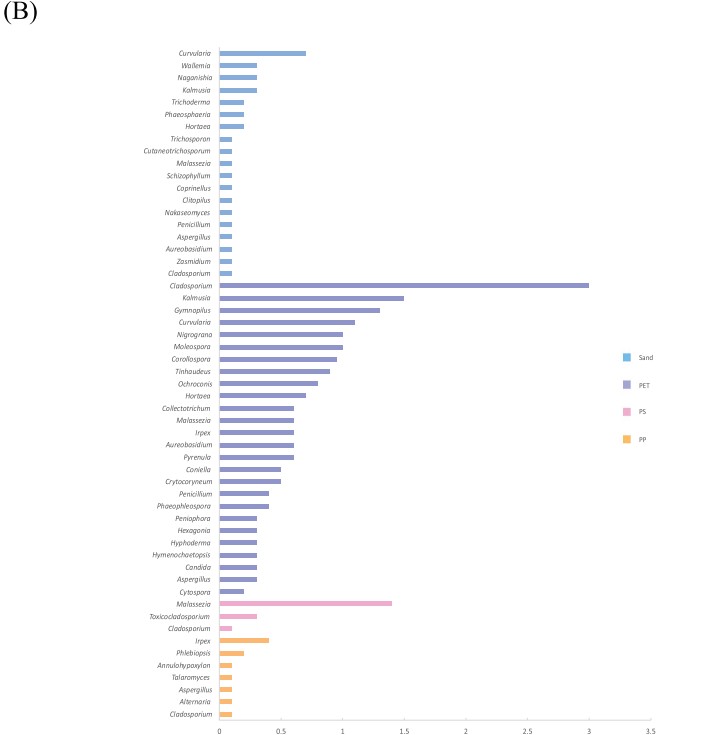

**Figure 8** **The dominance of bacterial and fungal genera in PP, PS, PET, and sand samples.** (A) Bacterial genera represented by LDA scores, and (B) Fungal genera represented by microbial composition bar plot.

(A)

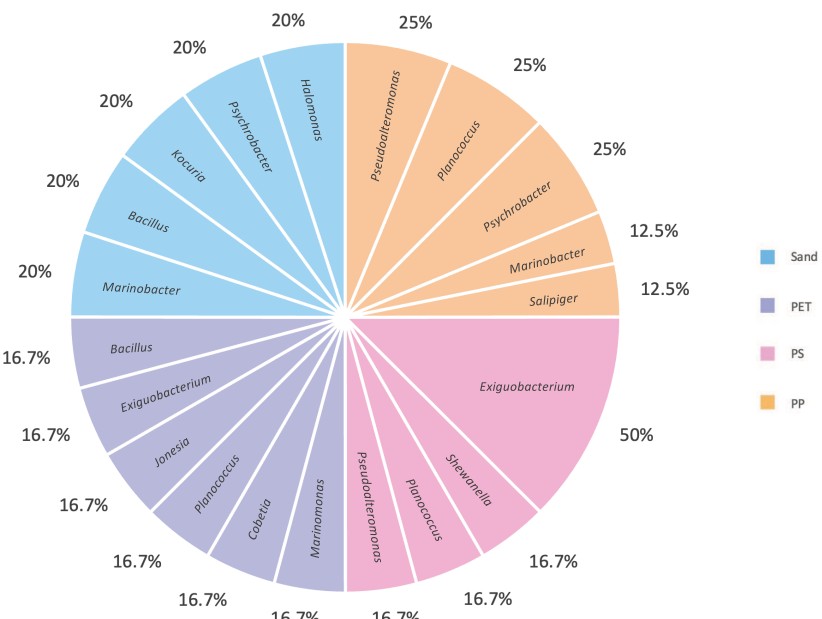

(B)

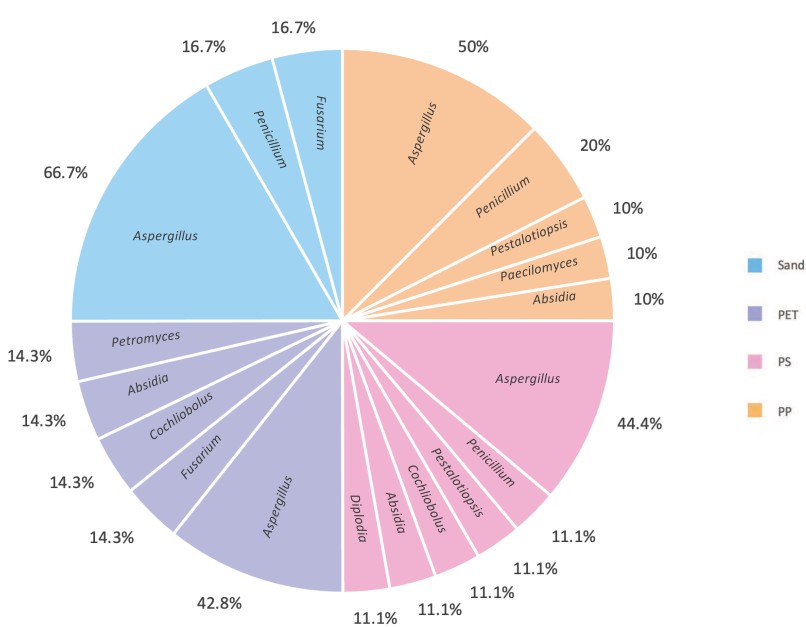

**Figure 9 Pie charts represent the microbial composition of bacterial and fungal genera in PP, PS, PET, and sand samples under culture-dependent condition.** (A) Bacteria and (B) fungi.

Chanthaburi, and Trat (*Bissen & Chawchai, 2020*). A previous report reveal the different plastic types were identified in marine environments, with PE and PS predominantly found on the sea surface and polyamide, PVC, and PET in the sediment (*Du et al., 2022*). The most common marine plastic debris in the marine environment was PE, followed by PP and PS (*Vaksmaa et al., 2021*). Analytical methods, such as Raman spectroscopy, atomic force microscopy, X-ray diffraction, and differential scanning calorimetry, can be used for plastic analysis. A Fourier Transform Infrared Spectrometer (FT-IR) is a popular technique for identifying functional groups within polymers, and in this study, ATR-FTIR was employed to analyze 16 plastic samples, revealing PP, PS, PET, HDPE, and an unidentified plastic. Furthermore, laboratory isolation revealed bacterial colonies ranging from $10^3$ to $10^7$ CFU/ml, whereas fungal colonies were $<10^2$ CFU/ml. These microorganisms are present in marine environments, with estimated abundances ranging from $10^3$ to $10^{10}$ cells/cm$^3$ in sediments and $10^4$ to $10^7$ cells/mL in seawater (*Wang et al., 2021*). Fifty-four bacteria and 62 fungi were successfully isolated from plastic and sand samples under laboratory conditions. However, despite the higher number of fungal isolates (62) than bacterial isolates (54), fungal diversity was lower than bacterial diversity.

The identification of bacteria through 16S rRNA revealed the several bacteria in the samples, including *Marinobacter*, *Pseudoalteromonas*, *Salipiger*, *Planococcus*, *Psychrobacter*, *Shewanella*, *Exiguobacterium*, *Marinomonas*, *Cobetia*, *Jonesia*, *Bacillus*, *Kocuria*, and *Halomonas*. For example, *Pseudoalteromonas* and *Psychrobacter* were found in the PP samples; these genera have previously been reported predominantly on PP, PE, and PET debris and are known hydrocarbon-degrading bacteria (*Basili et al., 2020*), *Exiguobacterium* was present in the PS samples, demonstrated the ability to form a biofilm on PS film during 28 days of incubation (*Ho, Roberts & Lucas, 2018*). *Salipiger* was found in PP sample, a member of the Rhodobacteraceae family, is commonly found on PP and PE plastic sheets (*Pinto et al., 2020*). Additionally, seven genera with the capability to degrade plastics were identified: *Salipiger*, *Planococcus*, *Psychrobacter*, *Shewanella*, *Jonesia*, *Bacillus*, and *Kocuria*. Notably, to our knowledge, this study represents the first report of *Planococcus* and *Jonesia* exhibiting BHET-degrading abilities. Moreover, *Shewanella* sp. exhibited a low similarity (<99%) comparing to the closet strain (*Shewanella baltica* NCTC 10735$^T$). *Shewanella* from deep sediment in the Kurile and Japan trenches has been reported to degrade PCL (*Urbanek, Rymowicz & Mirończuk, 2018*). These findings make these three genera particularly interesting for further study. Furthermore, the fungal ITS region revealed the presence of *Aspergillus*, *Penicillium*, *Pestalotiopsis*, *Peacilomyces*, *Cochliobolus*, *Absidia*, *Fusarium*, *Diplodia*, and *Petromyces*. *Aspergillus*, *Penicillium*, and *Fusarium* were present in both the plastic and sand samples, common fungi in marine environments (*Zeghal et al., 2021*). Additionally, five fungal genera exhibited plastic-degrading abilities: *Aspergillus*, *Penicillium*, *Peacilomyces*, *Absidia*, and *Cochliobolus*, previous reported to attach to the low-density PE (LDPE) surface (*Passos, Marconato & Franchetti, 2015*). Moreover, *Paecilomyces variotii* have been reported to have the ability to degrade PE (*Ali et al., 2021*). The most common fungal genera isolated under laboratory conditions were *Aspergillus* and *Penicillium*, extensively studied for their degradation properties. Various species of *Aspergillus* and *Penicillium* have demonstrated

the ability to degrade PE, LDPE, HDPE, and polyurethane (PU) (*Zeghal et al., 2021*). *Fusarium* has the capacity to degrade PET and PCL (*Fesseha & Abebe, 2019*). Moreover, *Aspergillus*, *Penicillium*, and *Fusarium* have been employed as biological agents for degrading PET and PS foam (*Srikanth et al., 2022*).

In the laboratory isolation, no microorganisms demonstrated the PS or PVC degradation on MSM-emulsified plastic agar. PS is known for its strong hydrophobic property and extremely stable polymers with high molecular weight, resists biodegradation (*Mohanan et al., 2020*). Despite this, some marine microbes capable of degrading PS have been identified in previous reports. For example, *Pseudomonas aeruginosa* has demonstrated the ability to degrade modified PS, *Curvularia* sp. isolated from soil samples could degrade atactic PS, and *Rhodococcus ruber* and *Enterobacter* sp. exhibit the ability to degrade three forms of PS in synthetic medium (*Mohanan et al., 2020*). Moreover, fungi such as *Aspergillus*, *Cladosporium*, and *Wallemia* isolated from the Western South Atlantic and Antarctic Peninsula, have shown an affinity for attaching to PS (*Du et al., 2022*). In addition, PVC, known for its low cost and resistance to biological and chemical factors, is a hard plastic with polymer structures composed of repeating chloroethyl units. PVC can exist in both rigid and flexible forms, being insoluble in alcohol but soluble in tetrahydrofuran. Compared to PET and PS, PVC is generally considered more challenging to biodegrade (*Srikanth et al., 2022*). Nevertheless, certain microbes, such as *Pseudomonas citronellolis* and *Bacillus flexus*, exhibit the ability to biodegrade PVC film, demonstrating substantial depolymerization activity towards PVC additives. *Pseudomonas putida* strain AJ has also been reported to utilize vinyl chloride monomers as a carbon source for growth (*Mohanan et al., 2020*). Moreover, the fungi *Phanerochaete chrysosporium* PV1 and *Aspergillus niger* PV3 isolated from PVC films buried in the soil, have demonstrated PVC degradation abilities (*Ali et al., 2021*).

Microbes involved in plastic degradation produce various hydrolytic enzymes, including cutinase, lipase, protease, esterase, laccase, or peroxidase (*Srikanth et al., 2022*). Bacterial enzymes such as cutinase and PETase have demonstrated efficacy in degrading PET and aliphatic polyesters like PCL (*Mohanan et al., 2020*). Arctic sea ice bacteria such as *Marinomonas*, *Pseudoalteromonas*, and *Pseudomonas* exhibit extracellular lipase activity capable of hydrolyzing polyesters like PCL (*Urbanek, Rymowicz & Mirończuk, 2018*). Laccase from *Rhodococcus ruber* C208 can degrade LDPE film, while the hydroquinone peroxidase from *Azotobacter beijerinckii* HM121 degrade PS film (*Ru, Huo & Yang, 2020*). Moreover, esterase from *Enterobacter* sp. HY1 exhibit the abilities to degrade BHET (*Qiu et al., 2020*). Other studies have identified fungal enzymes involved in plastic degradation. For example, cutinase from *Fusarium oxysporum* could degrade PET (*Zhou et al., 2022*), lipase B from *Candida antarctica* effectively hydrolyze PET into TPA, serine proteases from *Arthrobotrys oligospora* degrades PLA, and laccase from *Cochliobolus* sp. could degrade low molecular weight PVC (*Sumathi et al., 2016*). Esterase from *Curvularia senegalensis* has the capability to degrade poly (butylene succinate-co-adipate) and PU. Esterase from *Penicillium griseofulvum* could degrade PU. Moreover, manganese peroxidases and lignin peroxidases from *Aspergillus flavus*, *A. niger*, and

*Fusarium graminearum* could biodegrade PE carrier bags, with *A. niger* and *A. flavus* producing the most lignin peroxides (*Srikanth et al., 2022*).

This study conducted an analysis of the marine microbiome using 16S and ITS amplicon sequencing in both sand and plastisphere samples from Kung Wiman Beach. Following a review of pertinent literature on the marine plastisphere (*Du et al., 2022*) and marine microbial colonization in plastisphere (*Zhai, Zhang & Yu, 2023*), this is the first report on plastisphere investigation in Gulf of Thailand by field sampling. Recent reviews extensively covered various methodologies for studying plastisphere, encompassing field sampling, *in situ* field culture, and laboratory-simulated experiments (*Sun, Wo & Zang, 2023*). The application of omics technology would facilitate to elucidate diverse microbial ecology residing on plastisphere (*Tiwari et al., 2022*). These studied aimed to gain insight into the composition and dynamics of microbial communities associated with plastic debris in this specific region. Moreover, it is crucial to identify the mechanisms underlying plastic degradation process in order to accurately understand the natural occurrences in this regard (*Wright et al., 2020*). The bacterial microbiome at the phylum level revealed that Proteobacteria was the most abundant group in the PET plastisphere This phylum, along with Bacteroidetes, was more commonly found in plastic samples than in sand samples, consistent with previous studies indicating their common occurrence in marine environments. The microbial communities of plastics, seawater, and sediment were predominantly composed of Proteobacteria, Bacteroidetes, Actinobacteria, Cyanobacteria, Planctomycetes, and Firmicutes (*Basili et al., 2020*). Proteobacteria dominated, followed by Bacteroidetes and Cyanobacteria, in the plastisphere collected from the Mediterranean Sea (*Vaksmaa et al., 2021*). Similarly, in the North Pacific Subtropical Gyre, Proteobacteria, Bacteroidetes, and Cyanobacteria were the main phyla found on plastic particles (*Vaksmaa et al., 2022*). Moreover, Proteobacteria and Bacteroidetes were consistently present on plastic surfaces, even in deep water (*Du et al., 2022*). Furthermore, Cyanobacteria in these studied was more abundant in PS samples than in other samples. showing an affinity for the surface of micro- and nano-PS plastics (*De Oliveira et al., 2020*). Actinobacteria and Acidobacteria phyla were more abundant in sand samples than in plastic samples. For example, beach sand from Avalon (CA, USA) and Provincetown (MA, USA) exhibited major phyla, including Acidobacteria, Actinobacteria, Bacteroidetes, Proteobacteria, and Planctomycetes (*Halliday et al., 2014*). Regarding fungal community composition, the Ascomycota and Basidiomycota phyla were the most abundant groups in all samples, especially in PET samples, consistent with previous studies indicating their common occurrence in marine environments. A study in the Baltic Sea found that Ascomycota were highly abundant, followed by Basidiomycota and Mucoromycota (*Zeghal et al., 2021*). As 95.9% of all samples in this study could not be classified into specific groups by the Zymo Research database, a further fungal microbiome analysis was conducted upon comparison with the GenBank database, revealing that the proportion of other eukaryotic groups was higher than that of fungi. This finding is consistent with fungal community analyses conducted by Zymo Research Corporation. It is possible that the ITS primer pair (ITS3 forward and ITS4 reverse) targeting the ITS2 region was not specific enough in this study.

Comparing the number of microbial groups in microbiome analysis revealed greater diversity than that obtained through culture-dependent conditions. Specifically, microbiome analysis exhibited 15, 21, 15, and 22 bacterial genera, and seven, three, 26, and 19 fungal genera were identified from PP, PS, PET, and sand samples, respectively. In contrast, laboratory isolation demonstrated that five, four, six, and five bacterial genera, and five, six, four, and three fungal genera were isolated from PP, PS, PET, and sand samples, respectively. Furthermore, the bacterial species identified through LDA score, and the fungal species identified through microbial composition barplot, represent the most dominant microbial groups in each sample that were absent under culture-dependent conditions. For example, *Prochlorococcus* in the PP samples, *Ruegeria* in the PS samples, *Erythrobacter* in the PET samples, and *Rhodopirellula* in the sand samples. Additionally, the ITS microbiome indicated that *Cladosporium* and *Malassezia* were the most common fungal genera, while the fungi groups under laboratory conditions were mostly *Aspergillus* and *Penicillium*.

Several factors contribute to inability to cultivate marine microorganisms under laboratory conditions, such as unsuitable growth conditions, slow growth rates, differential culture media, and limited colony development (*Wang et al., 2021*). In this study, both isolation under laboratory conditions and microbiome analysis were employed to examine microbial groups, with some microbial groups showing consistency between these two approaches. The specific culture media and laboratory conditions influences the isolation of microorganisms, and certain microorganisms may not have been able to grow under these conditions, thereby limiting the assessment of their plastic degradation capabilities. These studied employed solid media (agar), specifically MA supplemented with cycloheximide for bacteria and PDA supplemented with nalidixic acid for fungi, to isolate microorganisms. Plastic degradation properties were tested on MSM agar supplemented with emulsified plastic. Similar approaches have been reported in other studies, bacterial strains were isolated from soil samples at the Rochester Institute of Technology (Rochester, NY, USA) cultured on Luria Broth (LB) (*Parthasarathy et al., 2022*), and arctic fungi and bacteria were isolated using Martin and LB agar plates, respectively. Additionally, a mineral minimum medium containing emulsified polymers was utilized to investigate plastics-degrading microorganisms (*Urbanek et al., 2017*). Moreover, there is a method available for testing plastic degradation using a liquid medium supplemented with plastic film. For instance, PLA-degrading strains from the GMKU culture collection were inoculated into a basal medium containing three PLA sheets (*Lohmaneeratana et al., 2020*), fungi was investigated using mineral minimum medium containing plastic film (*Urbanek et al., 2017*). Therefore, further studies can focus on plastic degradation in a liquid medium and the screening of enzymes with the capability to degrade plastics.

# CONCLUSIONS

Our study successfully isolated marine microorganisms (bacteria and fungi) from Kung Wiman Beach capable of degrading PLA and BHET. Notably, this study represents the first report of bacteria from the genera *Planococcus* and *Jonesia* with BHET-degrading properties. Furthermore, 16S and ITS microbiome analyses revealed dominant microbial

genera in the samples that were not identified under culture-dependent conditions. Future studies should focus on examining and confirming the biodegrading abilities of the isolated marine microorganisms and explore interesting genes or enzymes associated with their plastic degradation. These findings can potentially be applied to reduce microplastics in the environment, specifically in the Thai seas.

## ACKNOWLEDGEMENTS

We would like to express our gratitude to Asst. Prof. Dr. Sivawan Phoolphundh (Department of Microbiology, Faculty of Science, King Mongkut's University of Technology Thonburi) for kind suggestion and Dr. T. Pemberton for critical proofreading this manuscript.

### Funding

This research project was supported by Thailand Science Research and Innovation Basic Research Fund for 2021 (project number: FRB640008). The funders had no role in study design, data collection and analysis, decision to publish, or preparation of the manuscript.

### Grant Disclosures

The following grant information was disclosed by the authors:
Thailand Science Research and Innovation Basic Research Fund for 2021: FRB640008.

### Competing Interests

The authors declare that they have no competing interests.

### Author Contributions

- Nutsuda Chaimusik conceived and designed the experiments, performed the experiments, analyzed the data, prepared figures and/or tables, authored or reviewed drafts of the article, and approved the final draft.
- Natthaphong Sombuttra performed the experiments, analyzed the data, prepared figures and/or tables, authored or reviewed drafts of the article, and approved the final draft.
- Yeampon Nakaramontri performed the experiments, analyzed the data, prepared figures and/or tables, authored or reviewed drafts of the article, and approved the final draft.
- Penjai Sompongchaiyakul conceived and designed the experiments, prepared figures and/or tables, authored or reviewed drafts of the article, and approved the final draft.
- Chawalit Charoenpong performed the experiments, analyzed the data, prepared figures and/or tables, authored or reviewed drafts of the article, and approved the final draft.
- Bungonsiri Intra conceived and designed the experiments, analyzed the data, prepared figures and/or tables, authored or reviewed drafts of the article, and approved the final draft.
- Jirayut Euanorasetr conceived and designed the experiments, analyzed the data, prepared figures and/or tables, authored or reviewed drafts of the article, and approved the final draft.

## DNA Deposition

The following information was supplied regarding the deposition of DNA sequences:

The nucleotide sequences are available at GenBank: OR185572–OR185595 (bacteria) and OR205218–OR205249 (fungi).

## Data Availability

The 16S and ITS metagenome raw sequence reads from plastisphere and sand from Kung Wiman beach are available at NCBI PRJNA997562 and figshare: Euanorasetr, Jirayut (2023). 16S and ITS metagenomes raw data from plastisphere and sand at Kung Wiman beach. figshare. Dataset. https://doi.org/10.6084/m9.figshare.23733891.v1.

The FTIR spectra of plastic debris collected from Kug Wiman Beach is available at figshare: Euanorasetr, Jirayut (2023). FT-IR raw data from plastic debris at Kung Wiman beach. figshare. Dataset. https://doi.org/10.6084/m9.figshare.23797077.v1.

## Supplemental Information

Supplemental information for this article can be found online at http://dx.doi.org/10.7717/peerj.17165#supplemental-information.

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
