# Peer review of "The comparative plastisphere microbial community profile at Kung Wiman beach unveils potential plastic-specific degrading microorganisms"

_PeerJ, doi:10.7717/peerj.17165_

## Round 0.1 · original submission · Major Revisions

While all of the reviewers felt that the manuscript was well-written and of an interesting topic (as did I), there were sufficient concerns regarding details around the methods (in particular the 16S and ITS sequencing taxonomic analysis) that the manuscript requires major revisions.

Reviewer 1 ·

Basic reporting

The manuscript entitled “The comparative plastisphere microbial community profile at Kung Wiman Beach unveils potential plastic-specific degrading microorganisms” aimed at identifying microorganisms of plastic degradation potential using culture-dependent and culture-independent methods. Overall, the topic is interesting, and the English writing is fine. Figure qualities need to be improved as all are of low resolution at this time, and some of them can be combined (Please see my additional line-to-line comments).

Experimental design

The sampling strategy needs more details(Please see my additional line-to-line comments).

There are critical controls missing in the experiment design and/or sequencing data analysis. The culture-dependent experiment needs negative and positive controls. Also, it looks like all samples were taken from the Kung Wiman beach only. This may or may not impact the selective culture-dependent methods. However, for culture-independent methods (16S or ITS sequencing), how can the author make the correlation between the identified microorganisms and the plastic degradation potential? Additional data (e.g., sand) from a non-contaminated beach (or similar) as a negative control is needed to support the findings of this manuscript (Please see my additional line-to-line comments).

Validity of the findings

The 16S and ITS sequencing results need to be clearer.
The 95.9% unclassified fungi ITS data set needs validation before this manuscript can be published
Also, what is the hypothesis of the 16S and ITS sequencing analysis? The results also need better explanations (Please see my additional line-to-line comments).

Additional comments

Please see below my line-to-line comments.

Line 128: How were the plastic, sand, and seawater samples collected (i.e., how were they collected from the beach)? How many were collected for each sample type?

Line 133: How long were the water samples kept at room temperature? Also, what were the water samples used for i.e., they were not included in the results?

Line 159-176: Please consider these two paragraphs as one selective culture-dependent method section.

Line 170: What was the volume of microbial solution used for inoculation?

Line 179: Please describe the “microwave method”.

Line 190: Please specify the sequencing technique.

Line 241: Please provide information about the custom primers.

Line 272: From the supplemental materials, it looks like this is the content of organic carbon. Please clarify.

Line 274: For Figure S4, please consider adding proportional scales.

Line 287: This sentence is not clear. Were there 42 isolates or 27? Same as in line 290.

Line 294-297: This is methods. Please modify accordingly.

Line 302: Please consider moving tables 2 and 3 to supplemental materials.

Line 305: Controls are missing for this experiment. Did the author include isolate tests in MSM agar without plastics as a negative control? Did the author include known plastic-degrading bacteria/fungi in the same test as positive control(s)? How can readers be convinced that the clear zone is because of the microbial degradation of plastics?

Line 334: 95.9% of all samples unclassified is not a good sign. Did the author blast these unknown reads to see what they could be (e.g., on NCBI)? Could there be a classification error, or the custom ITS primer did not work well (not specific enough)?

Line 336-347: The Lefse analysis needs further explanation here. What do these identified species mean here? Did the author compare the microbial communities in the plastic samples to those in the sand samples? Are those species specific to (only present in) the sample type? How abundant are those species in the whole microbial community?

Also, are the sand samples in this beach also considered as “contaminated”? i.e., two sand species were also identified as “plastic degrading” in Fig. 3. In this case, the author will need to compare their contaminated samples with negative controls (e.g., clean sand samples from local sites, or publicly available clean beach samples) to support the correlation between their identified species and their potential to plastic degradation.

Please also consider combining figures 7/8 and figures 9/10, respectively.

Line 349: Discussion is overall too long. Please consider shorten this section.

Line 424-443: The mechanisms of enzymes in plastic degradation is interesting and could lead to the hypothesis of this study. Instead of leaving this detailed information in the discussion, I suggest moving them to the introduction to strengthen the manunscript.

Reviewer 2 ·

Basic reporting

This manuscript addressed an interesting topic in environmental microbiology. However, there are several areas within the manuscript that require attention before it can be considered for publication.

Experimental design

1. Inserting images into PDF/Word documents may reduce the resolution of figures. The images appear to be low-resolution, resulting in unclear visibility of many data points. Authors are advised to consult the staff to resolve this issue before publication. Additionally, the legends for the figures should provide more informative descriptions.
2. The authors used the term 'metagenomics analysis' throughout the manuscript, but they did not actually conduct metagenomic sequencing for analysis. This usage of the term 'metagenomics' in the manuscript can be misleading to readers.
3. A figure illustrating the study design and analysis would be helpful for better understanding.

Validity of the findings

1. Details should be provided for the 16S rRNA and ITS sequencing analyses, including the criteria and cut-offs used for adapter removal, error correction, and chimeric removal. Additionally, please provide the cut-offs used for LEfSe to ensure the reproducibility of the study.
2. Details on how the statistics were conducted, including the methods and values used, were not provided in the manuscript. For example, the authors should provide details on how the PcoA was conducted and the input data information for creating the figures.

Additional comments

Minor comments:
1. Statistical analysis of Shannon's indices for different groups should be conducted to determine whether there are significant differences.
2. Regarding the PCoA figures, the authors should describe how the dissimilarities of samples were calculated. Statistical analyses, such as Permutational Multivariate Analysis of Variance, should be conducted to assess whether significant differences exist among different groups.

Reviewer 3 ·

Basic reporting

In this manuscript, the authors aanalyzed microbiota of sand and plasticsphere from from KungWiman Beach and isolated and identifed bacteria and funfi that can degrade PLA and BHET. The manuscript is well-written, and the background information and references are adequately introduced and cited. Overall, the manuscript is highly intriguing and the findings hold great value for the field. I only have a few specific comments.
Abstract
1. L55-75: The authors need to concentrate the “Results” part to make it short and clear to follow.
2. L76: The authors may need to rewrite this sentence, similar to “We reported on the microbial communities found on the plastisphere at Kung Wiman Beach and isolated and identified microbes with the capacity to degrade PLA and BHET.”
Materials and Methods
3. L163: I am really concerned that the Ultrasonic cleaner for 5 min can kill some microorganisms, which may affect your final microbiota from the plasticsphere. Need references here and do the authors use this method to isolate bacteria and fungi before?
Results
4. For all main figures, authors need to merge some figures into one figure, like Fig. 2 and 3, Fig. 4, 5, and 6. Fig. 9 and Fig. 10 didn’t give enough information but they occupied a lot of space. Authors can consider changing them to some other figure forms or just tables.
For all supplementary figures, there is no figure title and figure fonts.
5. For all supplementary figures, there is no figure title and figure fonts.
6. L314: If authors have some other bacteria/fungi degradation figures of plastic, please include them. I think this part is interesting and will attract readers.
And do authors test the ability of plastic degradation of isolated bacteria/fungi in the broth media?
7. L342: “16S rRNA metagenomics analysis of plastic and sand samples revealed the dominant bacterial genera that were absent in the culture-dependent conditions....”
Need to give an explanation or hypothesis about why they are absent? Which kind of factors could lead to this issue?
Discussion
8. The Discussion section is lengthy and redundant (16 paragraphs, it’s too long), please condense it to make it easier to follow for readers.

Experimental design

See 1.

Validity of the findings

See 1

Additional comments

See 1

·

Basic reporting

Review for Chaimusik et al The comparative plastisphere microbial community profile at
Kung Wiman Beach unveils potential plastic-specific degrading microorganisms.

This study reports the degradation of various types of microplastics collected from the Kung wiman beach by both bacteria and fugal isolates from the surface of microplastics. this study further analyzed the composition of the plastisphere by metagenomics. Chaimusik et al have further reported the ability of Planococcus and Jonesia to degrade bis (2-hydroxyethyl) terephthalate (BHET) for the first time.

This is an interesting study that is elegantly written. I have a few minor concerns and the authors should address them to increase the strength of the manuscript.


Line 82 Please use the most recent data for plastic production from Plastics – the Facts
2022.

Line 94 Please add more recent citations like Haryati et al A review on microplastic
ingestion in marine invertebrates from Southeast Asia.
(DOI: 10.14456/sjst- psu.2022.84)

Line 133 Sea water was collected and stored in a gallon bottle- Please specify the
material in which the water is stored. If it’s a plastic bottle then it could
become a source of microplastic contamination.

Line 276 Please add the ATR-FTIR spectra of the polypropylene (PP), polystyrene (PS),
polyethylene terephthalate (PET), polyethylene (PE) as a control with various
plastics.

Line 313 Please provide the hydrolysis capacity (HC) values for the five fungal genera
(Aspergillus, Penicillium, Peacilomyces, Absidia, and Cochliobolus).


Line 314 Authors have provided only two figures with a zone of clearance on MSM
emulsified BHET agar for Planococcus and Absidia. Please provide the zone
of clearance figures with other bacteria and fungi with high HC values like
Salipiger sp. PP4 and Kocuria sp. Sand3.

Figure 2 Figure 2(A) shows the zone of clearance Planococcus sp. Why the color of
Agar looks different from Figure 2(B) and why the bacterial colony of
Planococcus looks like a heatmap or rainbow colored, is it processed in
ImageJ?

Figure S4 Please add a scale bar to the microplastics. A scale will help in
determining the exact size of these microplastics.

Figure S5 In all the ATR- FTIR please give the peaks related to the important bonds in
various functional groups.

Experimental design

no comment

Validity of the findings

no comment

Additional comments

Review for Chaimusik et al The comparative plastisphere microbial community profile at
Kung Wiman Beach unveils potential plastic-specific degrading microorganisms.

This study reports the degradation of various types of microplastics collected from the Kung wiman beach by both bacteria and fugal isolates from the surface of microplastics. this study further analyzed the composition of the plastisphere by metagenomics. Chaimusik et al have further reported the ability of Planococcus and Jonesia to degrade bis (2-hydroxyethyl) terephthalate (BHET) for the first time.

This is an interesting study that is elegantly written. I have a few minor concerns and the authors should address them to increase the strength of the manuscript.

---

## Round 0.2 · Minor Revisions

While all of the reviewers agreed that the manuscript has improved overall, each of them still had minor concerns that need to be addressed before the manuscript can be accepted for publication.

Reviewer 1 ·

Basic reporting

The revised manuscript has been improved in many ways.

Experimental design

Details of the collection method were not addressed. Did the author just pick up the plastic, or they were buried? Where did the author collect water samples? surface water? below surface? near-shore? Please provide these details.

Validity of the findings

The 95.9% non-classified fungi sequencing dataset is still concerning. The author claimed in the rebuttal letter that they did a re-analyzing of the data. However, no reanalysis was included in the revised results. The author did not provide information on what the unclassified reads could be. The reanalysis as a result validation should be included in the results to support the non-specificity of the primers, which should also be included in the results, not only in the discussion.

Reviewer 2 ·

Basic reporting

There are several areas that still need to be addressed before it can be accepted for publication.

1.The authors are required to deposit the raw reads in a public database and provide the accession number in a statement of data availability for readers to access the data.

Experimental design

There are two 'microbiome analyses' in the methods section. The authors are required to use more specific titles to distinguish them.

Validity of the findings

In response to the reviewers, the authors indicated that the bioinformatics analyses were conducted by Zymo Research Corporation. They denied providing details for the bioinformatics analyses, even though QIIME 2 and Dada2 are publicly available tools. The authors should be transparent in disclosing the use of Zymo Research Corporation for bioinformatics analyses in their manuscript.

Additional comments

NA

Reviewer 3 ·

Basic reporting

The authors have addressed my comments well. The figures resolution in this manuscript is too low to get enough information from figures. After increasing the figures' quality, the manuscript can be accepted.

Experimental design

As above

Validity of the findings

As above

Additional comments

As above

---

## Round 0.3 · accepted · Accept

The manuscript is now ready for publication, as the reviewers were happy that you have addressed their comments. There are two very minor comments from the reviewers (one a space that needs to be added after 'revealed' in the abstract, and the second noticed that the manuscript still refers to Fig. S9, but there no longer seems to be such a figure). Hopefully these can be fixed in post production tasks.

Reviewer 1 ·

Basic reporting

The manuscript is improved and addresses all the concerns I had. One more thing that needs to change for publication is the mentioned Fig. S9, as I did not find it in the submitted supplemental materials. However, even without this piece of data, the description of the BLAST analysis in the main text is enough. Please add Fig S9 or revise the text accordingly.

Experimental design

The manuscript is improved and addresses all the concerns I had.

Validity of the findings

The manuscript is improved and addresses all the concerns I had.

Reviewer 2 ·

Basic reporting

The manuscript has been improved significantly, and authors have addressed all of my comments.
In abstract, 114: a space should be added after ‘revealed’.

Experimental design

NA

Validity of the findings

NA

Additional comments

NA